# TRADING COMPLEXITY FOR EXPRESSIVITY: THEORETICAL EXPLORATION OF LINEAR & CAUSAL TOKEN MIXING STRATEGIES

## ABSTRACT

We revisit token mixing in sequence models through a unified, causal linear framework that separates two effects: (i) direct one-step influence of inputs on outputs and (ii) recurrent propagation of information through past outputs. This perspective encompasses major architectures – including attention, state-space models, and hybrids – while exposing simple design parameters that govern efficiency and expressivity. We show that every causal linear mixer can be written in this form, where computation reduces to solving a triangular system with well-understood numerical properties. The framework generalizes the recurrence equations of SSMs and linear attention by allowing each state to depend on multiple past states rather than only the immediate predecessor. This unlocks new tradeoffs between decoding speed, cache size, and ability to model long-range dependencies. Building on this view, we design structured recurrence patterns that provably achieve any desired complexity – trading runtime for expressivity in a principled way. Together, these results provide a unified toolkit for understanding and designing efficient, expressive token mixers across model families.

## 1  INTRODUCTION AND RELATED WORKS

Token mixing – the mechanism by which sequence models exchange information across positions – is the central design axis of modern architectures. Early recurrent neural networks (RNNs) modeled sequences by passing information forward one step at a time, but their inherent sequentiality limited parallelism and made long-range dependencies difficult to capture. Transformers replaced recurrence with content-based self-attention, enabling global one-hop interactions and massive parallelism, and quickly became the backbone of large language models (Vaswani et al., 2017; Devlin et al., 2019; Brown et al., 2020). Yet their quadratic cost in sequence length remains a bottleneck as context windows scale toward hundreds of thousands or even millions of tokens.

In response, the field has diversified into alternative mixers. Low-rank and kernelized forms of linear attention approximate the softmax operator while reducing cost (Katharopoulos et al., 2020; Choromanski et al., 2020; Wang et al., 2020). State space models (SSMs) reinterpret token mixing as structured linear recurrences, with certain formulations (Gu et al., 2022) reducible to fast convolutions and others designed for recurrent or scan-friendly execution (Gu & Dao, 2024; Dao & Gu, 2024a), achieving strong performance on long-range benchmarks. Increasingly, hybrid models combine these mechanisms – mixing SSMs with local or sparse attention, or alternating different mixer types across layers – to balance expressivity, efficiency, and cache size (De et al., 2024; Zancato et al., 2024; NVIDIA et al., 2025; Team et al., 2025).

This growing diversity underscores a key challenge: token mixing is no longer embodied by a single operator but by a toolbox of mechanisms, each making distinct trade-offs between complexity, expressivity, and execution mode. One underexplored but related dimension is the order of recurrence: whereas classical RNNs and most SSMs propagate information through a single previous state, higher-order recurrences allow dependence on multiple past states, offering richer expressivity at the cost of more complex updates. While this idea has received limited attention, a few notable efforts include log-linear attention (Guo et al., 2025), which induces a logarithmic-order recurrence, and ChaCAL (Fagnou et al., 2024), which formalizes an infinite-order recurrence.

In this work, we provide a unifying framework for token mixing. We show that every causal linear mixer can be decomposed into (i) direct one-step input influence and (ii) recurrent propagation through past outputs. This structured recurrence view covers attention, SSMs, and linear attention, while exposing how design parameters control complexity, cache size, and long-range capacity.

Our contributions are the following:

- We introduce a general framework for causal linear token mixing that captures attention, SSMs, and their hybrids as special cases.
- We provide theoretical insights into the trade-offs between computational complexity and expressive power.
- We construct token mixers spanning a controlled range of complexities, from $\mathcal{O}(n)$ and $\mathcal{O}(n \log n)$ up to $\mathcal{O}(n^{3/2})$ and $\mathcal{O}(n^2)$.
- We empirically validate these designs on synthetic benchmarks and language modeling pre-training tasks.

Taken together, our results offer a principled lens for analyzing and designing efficient, expressive token mixers, providing conceptual clarity across diverse architectures.

## 2 RELATED WORKS

**Attention and efficient variants.** Transformers popularized global attention, but its quadratic complexity has motivated numerous efficiency improvements. One line of work retains exact softmax attention while optimizing CUDA kernels (e.g., FlashAttention (Dao et al., 2022; Dao, 2023)). Another line alters the operator itself: sparse and local attention restrict interactions while preserving long-range connectivity (Child et al., 2019; Beltagy et al., 2020; Zaheer et al., 2020), while low-rank or kernelized linear attention formulations reduce complexity via feature maps or projections (Katharopoulos et al., 2020; Choromanski et al., 2020; Wang et al., 2020; Xiong et al., 2021).

**State Space Models (SSMs).** An alternative approach frames token mixing as a linear dynamical system. HiPPO-based methods project sequences onto orthogonal polynomial bases to retain long-range history (Gu et al., 2020), while S4 and successors use diagonal-structured operators that can be implemented efficiently, either as convolutions or through parallel scan algorithms (Gu et al., 2022; Smith et al., 2022). Adaptive variants, such as Mamba (Gu & Dao, 2024), introduce input-dependent gating to handle more complex sequence patterns, and Mamba-2 (Dao & Gu, 2024a) streamlines the recurrence while providing theoretical connections to linear attention: its structured recurrence is equivalent to a 1-semiseparable transformation matrix, linking SSMs with masked linear attention and other gated linear attention variants.

**Theoretical limitations of SSMs and Transformers.** SSMs provide efficient linear recurrences, but their memory of past inputs decays exponentially with distance (Wang et al., 2025b), limiting long-range dependencies. Transformers, in contrast, can attend globally but lack recurrence, making tasks like entity tracking or copying challenging (Jelassi et al., 2024; Fagnou et al., 2024), and hindering generalization to longer sequences than seen in training (Beck et al., 2024).

**Hybrids.** Many contemporary models combine mixers to exploit complementary strengths. For example, Griffin interleaves gated linear recurrence with local attention (De et al., 2024), and B'MOJO integrates SSMs, local, and sparse attention in a single layer (Zancato et al., 2024). Larger systems, such as Nemotron-H and Gemma 3, mix SSM and attention layers across the network to balance efficiency and expressivity (NVIDIA et al., 2025; Team et al., 2025). Other works systematically explore combinations of attention and SSM-like operators (Waleffe et al., 2024; Wang et al., 2025a; Arora et al., 2024b; Thomas et al., 2025). These hybrid designs motivate frameworks that can describe multiple token mixing mechanisms within a single mathematical form.

**Higher-order recurrence.** Beyond first-order recurrences, a few works explore multi-step or infinite-order dependencies. Higher-order RNNs were studied classically (Hush et al., 1991; Soltani & Jiang, 2017). More recently, log-linear attention exhibits a logarithmic-order recurrence (Guo et al., 2025), while ChaCAL implements an infinite-order recurrence with explicit causal structure (Fagnou et al., 2024). These examples motivate our generalization of recurrence patterns beyond the standard first-order view.

## 3 FRAMEWORK

We cast token mixing as a *causal* linear operator that cleanly separates (i) direct, single-hop contributions from inputs and (ii) recursive, multi-hop contributions propagated through past outputs.

### 3.1 RECURRENCE AND MATRIX FORMS

For token embeddings $\{x_i\}_{i=1}^L$ and outputs $\{y_i\}_{i=1}^L$, we write the per-position recurrence

$$y_i = \underbrace{\sum_{j=1}^{i} \alpha_{ij}\, x_j}_{\text{present/past inputs (direct mixing)}} + \underbrace{\sum_{j=1}^{i-1} \beta_{ij}\, y_j}_{\text{past outputs (recurrent mixing)}} . \tag{1}$$

Here $\alpha_{ij} = 0$ for $j > i$ and $\beta_{ij} = 0$ for $j \geq i$ enforce causality. Stacking coefficients into lower-triangular matrices $A = [\alpha_{ij}]$, $B = [\beta_{ij}]$, and stacking tokens as $\boldsymbol{x} = [x_1^\top \cdots x_L^\top]^\top$, $\boldsymbol{y} = [y_1^\top \cdots y_L^\top]^\top$, yields the compact form

$$\boldsymbol{y} = A\,\boldsymbol{x} + B\,\boldsymbol{y} \iff \boldsymbol{y} = (I - B)^{-1} A\,\boldsymbol{x}, \tag{2}$$

which we refer to as the *general recurrent token mixing framework*. When $B$ is strictly lower-triangular (zero diagonal), $(I - B)$ is always invertible and $(I - B)^{-1} = \sum_{k=0}^{L-1} B^k$ (finite-depth solve). Expanding $(I - B)^{-1}A = A + BA + B^2 A + \cdots$ reveals a path-sum interpretation: one first attaches to inputs through $A$, then accumulates multi-hop contributions via $B$.

### 3.2 RELATION TO PRIOR UNIFYING VIEWS

Our formulation relates to recent efforts to unify attention and state-space models (SSMs) while remaining distinct. Most relevant is the *Structured State-Space Duality (SSD)* of Dao & Gu (2024b), which ties SSM sequence maps to (sequentially) semiseparable matrix mixers and shows that certain SSMs admit both linear (recurrent) and quadratic (attention-like) evaluation via structured masked attention (SMA). In SSD, attention appears as a quadratic form and efficiency comes from block decompositions exploiting low-rank off-diagonal structure, motivating Mamba-2 and its SSD kernel.

Methodologically, SSD studies the global transformation $T$ (its semiseparable structure, block low-rank properties, fast matvecs). We instead factor $T = (I - B)^{-1}A$, separating a direct channel $A$ (one-hop input mixing) from a recursive channel $B$ (multi-hop output coupling). Working at the level of $B$ gives precise control over causality, locality, and path length: sparsity in $B$ (as studied in Section 4) directly determines recurrent receptive-field, while $A$ independently sets instantaneous mixing. This primitives-first view defines a design space over lower-triangular pairs $(A, B)$, rather than properties of a fixed global $T$.

Unlike SSD, we do not assume semiseparable structure. Varying sparsity in $B$ and patterns in $A$ smoothly spans attention ($B{=}0$), many recurrent and SSM layers, and rational/resolvent mechanisms within a single recurrence/matrix statement. A detailed discussion of the examples encompassed by the framework is given in Appendix B. SSD's duality is recovered when its structural assumptions hold while our analysis and design rules apply more broadly.

SSD's architectural choices instantiate specific neighborhoods in our template: diagonal $A$, strictly lower $B$, factorizations of $(I{-}B)^{-1}$, and structured masks correspond to constrained direct/recursive channels. We thus provide an operator-level framing that puts attention and SSMs on equal footing, with minimal algebra that subsumes SSD's equivalences and adds interpretability and controllability via the structured recurrence $B$.

## 4 PATTERN DESIGN

An advantage of representing token mixing with Equation 2 is that we can enforce structure on $A$ and $B$ while still maintaining good expressivity. Indeed even if $B$ is very sparse (e.g. diagonal SSMs), the inverse $(I - B)^{-1}$ will be a dense lower-triangular matrix which may model complex behaviors. In this section we explore how the structure of $A$ and $B$ influence time and memory complexity, as well as measures of expressivity. All proofs can be found in the Appendix.

## 4.1 TRANSLATION INVARIANT PATTERNS

We start by investigating the class of attention patterns that are invariant by translation. They come as a simple choice, as they can be represented by a single strictly increasing function $f : \mathbb{N} \to \mathbb{N}^*$, such that the token at position $t$ will pay attention to tokens at positions $t - f(0), t - f(1), \ldots$

**Proposition 4.1** (Time complexity). *The token mixing layer induced by $f$ has a time complexity in $\mathcal{O}(g(n))$ for decoding the $n$-th token, where: $g(n) = \max\{i \in \mathbb{N} \text{ s.t. } f(i) < n\}$.*

*If $f$ can be extended to an invertible real function $\mathbb{R} \to [1, \infty)$ then the complexity is in $\mathcal{O}(f^{-1}(n))$. We will assume this is the case in the rest of the paper for simplicity.*

For example we may choose $f$ to be linear, quadratic or exponential, which will imply a time complexity respectively linear, square-root, and logarithmic.

In addition, we may want $f$ to satisfy a few other properties:

- $f(0) = 1$ ensures that all tokens have access to the previous token, and guarantees at least the same expressivity as a diagonal SSM,

- $f$ being convex, which means the attention pattern gets sparser and sparser the further we are from the current token. This makes sense since closer tokens are usually more important than those far away.

### 4.1.1 SHORTEST INFORMATION PATH

One metric for measuring expressivity in such models is the shortest path that information can follow from a token $i$ to a token $j > i$. Indeed, while in an attention layer all tokens are directly connected (distance of 1), recurrent models struggle at capturing long-range dependencies (Wang et al., 2025b), since information has to be stored in memory for a long period of time. The longer the information path is, the harder it is to learn, especially because of vanishing gradient effects.

**Proposition 4.2** (Shortest path). *Given two positions $i < j$, the length of the shortest path from token $i$ to token $j$ is:*

$$d(i,j) = \min\left\{ d \in \mathbb{N} \text{ s.t. } \exists\, a \in \mathbb{N}^d, \sum_{k=1}^{d} f(a_k) = j - i \right\} \tag{3}$$

*That is, the length depends on how many values of $f$ are needed to decompose the integer $j - i$.*

**Corollary 4.3.** *While Equation 3 is a complex problem to solve, simple choices for $f$ lead to closed-form solutions:*

- *If $f(i) = 2^i$, then $d(i,j)$ is the number of ones in the binary representation of $j - i$. This gives the bound $d(i,j) \leq \log_2(j-i)$.*

- *If $f(i) = i^2 + 1$, then by Lagrange's four-square theorem, we find that $d(i,j) \leq 4$.*

### 4.1.2 CONGESTION

A major problem of standard recurrent models is that all past information is compressed into a single vector, which makes it impossible to recall large pieces of information (Jelassi et al., 2024). By introducing additional connections to older hidden states, we aim at alleviating this bottleneck.

We formalize this via *graph congestion*, in a setup similar to Jelassi et al. (2024). The model is tasked with copying a sequence of length $n$ from input positions $1, \ldots, n$ to output positions $n+1, \ldots, 2n$. Consider a single token-mixing layer that defines a directed graph $\mathcal{G}$ over these $2n$ nodes according to its update patterns ($A$ and $B$).

Let $\mathcal{P}$ be a set of $n$ directed paths in $\mathcal{G}$, where the $i$-th path connects input node $i$ to output node $i + n$. We define the *congestion* of the layer as the maximum number of paths passing through any single node:

$$C(\mathcal{G}) := \max_{1 \leq i \leq 2n} \#\{\, p \in \mathcal{P} \mid i \in p \,\}. \tag{4}$$

Intuitively, $C(\mathcal{G})$ measures the largest number of information flows that must pass through a single hidden state. Standard recurrent models induce high congestion, since all paths must pass through the single hidden vector, whereas higher-order recurrences can reduce $C(\mathcal{G})$ by distributing information across multiple states.

**Proposition 4.4** (Lower bound on congestion)**.** *If we know that the shortest path between token $i$ and $i + n$ is at least $d$ long, for all $1 \leq i \leq n$, then we get:*

$$C(\mathcal{G}) \geq \frac{d + 1}{2} \tag{5}$$

**Proposition 4.5** (Upper bound on congestion)**.** *If the pattern is translation invariant, and we know that the shortest path between token $i$ and $i + n$ is at most $D$ long, for all $1 \leq i \leq n$, then we get:*

$$C(\mathcal{G}) \leq D \tag{6}$$

**Corollary 4.6.** *Combining Corollary 4.3 with Proposition 4.5, we get that:*

- *If $f(i) = 2^i$, then $C(\mathcal{G}) \leq \log_2(n)$.*

- *If $f(i) = i^2 + 1$, then $C(\mathcal{G}) \leq 4$.*

Together, Propositions 4.4 and 4.5 suggest a direct link between shortest information path and congestion.

Table 1: Properties of token mixing strategies using different structures.

| Structure | Time per token | KV cache size | Shortest path between tokens | Max congestion |
|---|:---:|:---:|:---:|:---:|
| Attention | $\mathcal{O}(n)$ | $\mathcal{O}(n)$ | 1 | 1 |
| Local attention | $\mathcal{O}(k)$ | $\mathcal{O}(k)$ | $\infty$ | 1 |
| Diagonal SSM | $\mathcal{O}(1)$ | $\mathcal{O}(1)$ | $n$ | $n$ |
| Local recurrence | $\mathcal{O}(k)$ | $\mathcal{O}(k)$ | $\frac{n}{k}$ | $\frac{n}{k}$ |
| General | $\mathcal{O}(n)$ | $\mathcal{O}(n)$ | 1 | 1 |
| $f(i) = 2^i$ | $\mathcal{O}(\log_2 n)$ | $\mathcal{O}(n)$ | $\log_2 n$ | $\leq \log_2 n$ |
| + cache-efficient | $\mathcal{O}(\log_2 n)$ | $\mathcal{O}(\log_2 n)$ | | |
| $f(i) = i^2 + 1$ | $\mathcal{O}(\sqrt{n})$ | $\mathcal{O}(n)$ | 4 | $\leq 4$ |
| + cache-efficient | $\mathcal{O}(\sqrt{n})$ | $\mathcal{O}(\sqrt{n})$ | | |

## 4.2 CACHE-EFFICIENT TRANSFORMATION

One drawback of the patterns considered above is that the KV-cache size remains in $\mathcal{O}(n)$ despite the time complexity being much better. This is due to the past indices being offset by 1 at each step. We propose a simple algorithm for generating cache-efficient patterns.

The idea is that at time $t$ we only reuse indices from time $t - 1$. Instead of attending to the past index $j_t(i) = t - f(i)$ for $i \in \mathbb{N}$, we look for the closest $j' \geq j_t(i)$ that was used at time $t - 1$. This ensures that any token attended was already in the cache. Surprisingly, this leads to a very structured pattern.

**Proposition 4.7.** *Using the cache-efficient pattern induced by the function $f$, the token at time $t$ will attend to the positions $p_t(i)$ for $i \in \mathbb{N}$ (with $f(i) < t$) with:*

$$p_t(i) = a_i \left\lfloor \frac{t - f(i)}{a_i} \right\rfloor + (a_i - 1) \tag{7}$$

*where $a_0 = 1$ and $a_{i+1} = a_i \left\lceil \frac{f(i+1) - f(i)}{a_i} \right\rceil$.*

We show visualizations of such patterns in Figure 1. We observe a periodic structure, where the indices associated with $f(i)$ increase by $a_i$ every $a_i$ timesteps.

**Proposition 4.8.** *The cache-efficient version has a decoding time and cache size both in $\mathcal{O}(f^{-1}(n))$.*

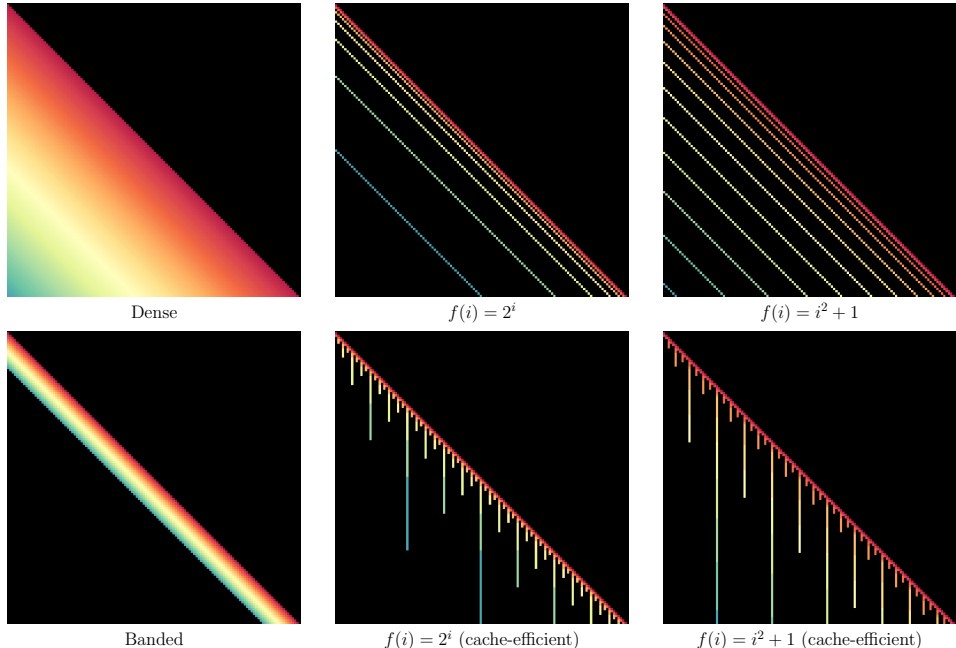

Figure 1: Illustration of different matrix structures, where non-zero entries are colored. The time and space complexity of the layer depends on the properties of the sparsity pattern.

## 5 EXPERIMENTS

We validate our claims with two sets of experiments. First, we use synthetic tasks to isolate and probe specific capabilities of token-mixing layers under controlled conditions. These tasks are designed to stress the theoretical knobs introduced in our framework (path length, congestion, cache structure). Second, we assess end-to-end performance on real-world data by training language models on OpenWebText. Together, the results test whether the theoretical predictions survive contact with training dynamics and natural language statistics.

### 5.1 MODELS

We base our architecture on the standard transformer, and in particular on GPT-2 (Radford et al., 2019), with the exception that we use RoPE positional embedding (Su et al., 2024) instead. The attention layer is swapped for one of the token-mixing layers studied in this paper, all implemented within the same backbone (same depth/width schedule, normalization, MLP blocks) so that comparisons isolate the effect of token mixing. For reference, we include full attention and local attention with window size $w=8$ as baselines.

Within our framework, we compare several $(A, B)$ structures (shared sparsity for $A$ and $B$ unless otherwise noted): *dense* (lower-triangular) $A$ with strictly lower-triangular $B$ (full resolvent mixer), *banded* with bandwidth $w=8$, and two translation-invariant families controlled by stride functions $f$: exponential $f(i)=2^i$ and quadratic $f(i)=i^2+1$. When relevant, we also evaluate cache-efficient variants (Section 4.2), which sparsify the working set while preserving the induced access pattern. Practical aspects (normalization of $A$ and $B$, conditioning of $(I-B)$, and implementation details) are discussed in Appendix A.

### 5.1.1 SETUP

We consider three canonical sequence problems adapted from prior work, chosen to stress different aspects of path geometry and memory pressure:

1. **Copy**: The model must copy an input sequence of size $L$ (Arjovsky et al., 2016; Jelassi et al., 2024). This task measures the ability of the model to memorize the sequence, and directly measures the congestion in the token mixing layers.

2. **Associative recall**: A similar yet more challenging task, where the model is given a series of key-value pairs that it must memorize. Then, when queried the keys, it must output the corresponding values. This measures whether the mixer can maintain a structured, addressable memory and isoften used to benchmark SSM-like models (Arora et al., 2024a; Dao & Gu, 2024a).

3. **Multi-hop recall**: Inspired from Fagnou et al. (2024), this task requires the model to solve a chain of associative recalls, which is particularly difficult for non-recurrent models. We modify the associative recall task by replacing some values by keys, that the model must then recursively lookup. This measures the state-tracking ability of the models.

To avoid overfitting to a single length scale, we randomize sequence lengths per batch. All models consist of two Transformer blocks; the first serves to preprocess the token stream into a representation amenable to the mixer, and the second performs the task-specific transport. Training, optimization, and sampling protocols are kept identical across models. Full hyperparameters and setup details are given in Appendix C.

### 5.1.2 RESULTS

We report the results for the synthetic tasks in Table 5.1.2.

Only the most general formulation with $A$ and $B$ dense is able to perfectly solve all tasks. While standard attention works as well for the copy and associative recall tasks, it struggles on multi-hop recall, which is expected (Fagnou et al., 2024). Local attention performs poorly across all settings.

The banded structure performs relatively well but falls behind patterns that involve more long-distance connections. Between $f(i) = 2^i$ and $f(i) = i^2 + 1$, the difference is slim but the exponential pattern is slightly worse, especially on the copy task.

The results appear especially encouraging for the cache-efficient variants. They even outperform their original counterparts, which is surprising. Still, this suggests that sparsifying the cache does not reduce the expressivity of the layer.

Table 2: Accuracy of diverse token mixing layers on the synthetic tasks. Results are averaged over 3 runs.

| Model | Copy | Associative recall | Multi-hop recall |
|---|---|---|---|
| Attention | **99.79**% $\pm$ 0.09 | **100.00**% $\pm$ 0.0 | 39.21% $\pm$ 1.54 |
| Local attention ($w = 8$) | 21.41% $\pm$ 0.06 | 26.20% $\pm$ 0.05 | 23.59% $\pm$ 0.59 |
| Dense $A$ and $B$ | **100.00**% $\pm$ 0.0 | **99.99**% $\pm$ 0.00 | **99.80**% $\pm$ 0.07 |
| Banded ($w = 8$) | 48.96% $\pm$ 1.93 | 41.12% $\pm$ 9.65 | 39.08% $\pm$ 1.50 |
| $f(i) = 2^i$ | 52.03% $\pm$ 0.21 | 49.03% $\pm$ 0.21 | 34.85% $\pm$ 0.11 |
| + cache-efficient | 60.99% $\pm$ 2.56 | 52.59% $\pm$ 1.79 | 38.63% $\pm$ 0.62 |
| $f(i) = i^2 + 1$ | 59.88% $\pm$ 1.13 | 53.61% $\pm$ 1.10 | 35.68% $\pm$ 0.33 |
| + cache-efficient | 60.31% $\pm$ 4.90 | 54.56% $\pm$ 1.43 | 38.02% $\pm$ 1.27 |

## 5.2 LANGUAGE MODELING

In this section we evaluate the models on a language modeling task using the OpenWebText dataset (Gokaslan & Cohen, 2019). The goal is to confirm that the theoretical insights, and the results on synthetic tasks, can transfer to real natural language. We train small transformers with 6 layers, while replacing attention with various token mixing layers. The models are trained for 100k steps with a context size of 512. The full experimental setup is detailed in Appendix C.

**Results Analysis**    Here, we analyze the results presented in Figure 2, first by comparing results inside each class and then comparing every trained models' performance.

**Comparison *inside* each class.**

1. $O(n)$ *time.* Within the full-complexity regime, the resolvent mixer $(I-B)^{-1}A$ with dense lower-triangular $A$ and strictly lower-triangular $B$ consistently sits below the standard full-attention cluster, achieving lower perplexity at comparable per-token complexity. This advantage is obtained with only a modest parameter increase (about $+7\%$ in our configurations), pointing to a *structural* benefit rather than a brute-force capacity effect. Intuitively, strictly lower-triangular $B$ accumulates causal summaries; the resolvent $(I-B)^{-1}$ expands them into a dense, geometry-aware receptive field; $A$ then reprojects, yielding more effective long-range mixing than dot-product attention for the same compute class.

2. *Sublinear variants $(O(\sqrt{n})$–$O(\log n))$.* First, we observe that the $O(\sqrt{n})$ model gives the better results in this bracket, corroborating with its denser $B$ matrix. Among structured sublinear designs, we observe a tight low-perplexity frontier, with two distinct behaviors. The strongest $O(\sqrt{n})$ instance beats by more than a point its cache-efficient variant, while the other sublinear propositions give performance very close to the ones of their cache-efficient versions. Lastly, in this setting, it is interesting to note that the standard attention-based model (local attention) is largely beaten (almost 10 PPL points) by its recurrent counterparts in $\mathcal{O}(k)$.

3. *Recurrent $O(1)$.* Constant-time models cluster at higher perplexities with comparatively small spread. This aligns with the congestion view: compressing the entire past into a single evolving state under-exploits long-range dependencies at a 512-token context under the given budget. This proposition still however beats the local attention method by around 8 PPL point.

**Comparison *across* classes: the Pareto frontier.**    Taken together, the points trace a clear speed–accuracy Pareto curve. Large gains occur when moving off $O(1)$ to sublinear access; by $O(\log n)$, diminishing returns appear, with several cache-efficient models matching the attention band – thus recovering most of attention's accuracy at substantially lower theoretical cost. If $O(n)$ is affordable, the resolvent mixer "wins the bracket," surpassing standard attention without a material parameter increase; if not, well-designed $O(\sqrt{n})$ or $O(\log n)$ caches provide near-attention perplexity at a fraction of per-token complexity.

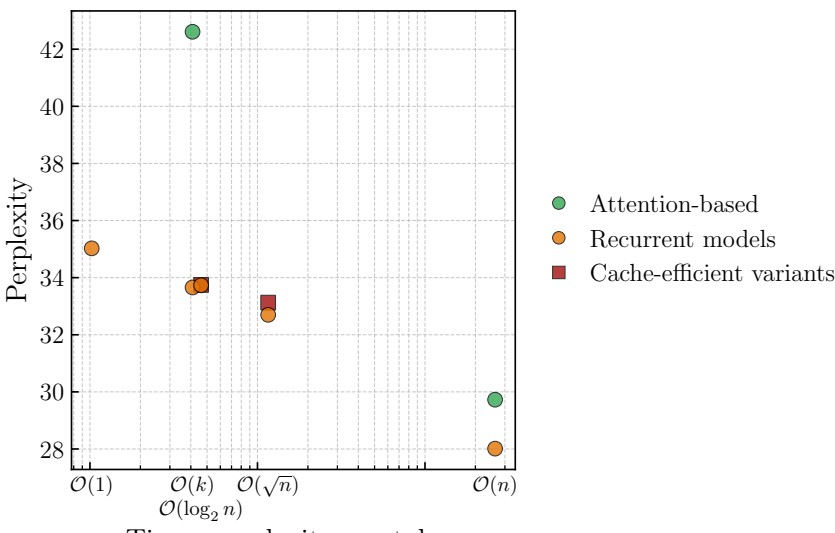

Figure 2: OpenWebText perplexity (lower is better) versus per-token time complexity for all trained 6-layer models. Points correspond to attention-based, recurrent, and cache-efficient variants; the $x$-axis annotates canonical regimes $O(1)$, $O(k)$, $O(\log_2 n)$, $O(\sqrt{n})$, and $O(n)$.

## 6    DISCUSSION

Our factorization $y = (I - B)^{-1}Ax$ gives a primitives-first view of causal token mixing where $A$ is an instantaneous input channel and $B$ a recursive multi-hop channel. Varying their sparsity and structure continuously spans families often treated as disjoint, while keeping triangular solves and strict causality explicit. The translation-invariant constructions parameterized by a strictly increasing $f$ make complexity and geometry analyzable: decoding cost $O(f^{-1}(n))$, shortest-path lengths tied to integer decompositions of $j-i$, and congestion bounds that diagnose memory bottlenecks. Choices like $f(i)=2^i$ (log-time, logarithmic depth) or $f(i)=i^2+1$ (square-root time, constant depth) demonstrate that "global vs. local" is not a binary; it is a tunable Pareto surface where hop depth, connectivity, and cache size are jointly controlled by $(A, B)$ and the pattern $f$.

Empirically, this lens explains and organizes results across complexity "brackets." Cache-efficient variants – attending only to indices already present in the previous step's cache – retain the same $O(f^{-1}(n))$ budget for *both* time and memory and often match (or only slightly degrade over) their non-cache counterparts, suggesting that structured $B$ acts as a useful inductive bias rather than a restriction. Aggregating models yields a clear speed–accuracy frontier: moving from $O(1)$ to sublinear access brings large gains; by $O(\log n)$, the proposed designs close most of the gap to attention. If $O(n)$ is permissible, the fully general resolvent mixer $(I-B)^{-1}A$ with dense lower-triangular $A$ and strictly lower-triangular $B$ clearly outperforms standard full attention at comparable per-token cost with only a marginal parameter increase. Practically, this suggests a design recipe: (i) select the target complexity class from systems constraints, (ii) pick $f$ to set hop geometry and congestion, (iii) enforce cache efficiency for memory locality, and (iv) tailor $A$'s parameterization to balance capacity and stability.

Limitations and systems implications follow directly. Our experiments use small models and unoptimized kernels; realizing end-to-end speedups requires specialized block-triangular forward-substitution kernels with regular memory access that exploit the periodic structure of cache-efficient patterns. While we discuss several practical considerations in Appendix A, there remain challenges to solve to make such token mixing layers scale efficiently and compete against other models. Stability and optimization might depend on parameterizations that preserve simple invariants ; alternative parameterizations could change training dynamics. Scaling studies at longer contexts and larger model sizes, heterogeneous layers mixing different $(A, B)$ structures, principled initializations for $B$ (for example inspired by the works on structured initializations for SSM (Gu et al., 2021)), and parameter-efficient factorizations for $A$ and $B$ (low-rank, semiseparable, or Toeplitz) are natural next steps to test whether the observed Pareto frontier persists at LLM scale.

## 7    CONCLUSION

This work reframes causal token mixing as a small set of explicit, controllable design choices, turning "which architecture?" into "which geometry and budget?". Rather than reiterating the factorization or translation-invariant analysis, we emphasize the shift in practice: pick a target complexity class, shape hop geometry to manage path lengths and congestion, and deploy cache-aware implementations that honor those choices. Our experiments – both synthetic probes and OpenWebText – anchor the theory by showing that sublinear access captures most of the accuracy of dense mixing at far lower budget, and that within the $\mathcal{O}(n)$ bracket a resolvent-style mixer can outperform standard attention with only marginal parameter overhead.

Looking forward, two tracks are most promising. On the *systems* side, specialized triangular-solve kernels and cache layouts are the lever to translate asymptotic gains into latency/throughput improvements at long context. On the *modeling* side, relaxing strict translation invariance toward learned, data-dependent patterns while preserving analyzable path and cost guarantees could broaden the design space without losing clarity. Our aim is not to crown a single mixer, but to provide a principled toolkit – geometry, capacity, and conditioning knobs – through which future architectures can be designed, analyzed, and engineered coherently.

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

# A  PRACTICAL CONSIDERATIONS

## A.1  PARAMETRIZATION OF $A$ AND $B$

Our framework does not make any assumption on how the coefficients $\alpha_{ij}$ and $\beta_{ij}$ are computed. That is, they could be constant (Katharopoulos et al., 2020), depend on the input (Dao & Gu, 2024a; Yang et al., 2023) or not (Gu et al., 2022), the distance (Sun et al., 2023), etc.

In an effort to bridge the gap between attention and SSMs, in all our experiments we choose attention-like coefficients for both $\alpha_{ij}$ and $\beta_{ij}$. That is, $A$ and $B$ are computed like two independent attention matrices, only with a modified normalization scheme (see Section A.2). We find this to be the most general way of parameterizing them, however more parameter-efficient choices should be considered in future work and may improve the overall efficiency.

## A.2  NORMALIZATION

A recurring problem in recurrent models is vanishing or exploding gradient. To prevent such phenomenon, one ought to carefully normalize the weights in $A$ and $B$. This is key for allowing the model to learn meaningful representations.

From Equation 1, we can see that if we assume that all $||x_j|| \leq C$, and $||y_j|| \leq C$ for $j < i$, we get:

$$||y_i|| \leq C \left[ \sum_{j=1}^{i} \alpha_{ij} + \sum_{j=1}^{i-1} \beta_{ij} \right] \tag{8}$$

We then only need to ensure that $\sum_{j=1}^{i} \alpha_{ij} + \sum_{j=1}^{i-1} \beta_{ij} = 1$, or in matrix notations: $(A + B)\mathbf{1} = \mathbf{1}$. In practice this can be done by modifying slightly the softmax equation. Note that this choice is similar to the gating mechanisms in GRUs for instance, while also matching exactly the attention softmax when $B = 0$.

However, one could also apply a softmax normalization to the full transformation matrix $T = (I - B)^{-1}A$. As discussed in Dao & Gu (2024a), this can be achieved by computing the forward pass $Tx$ without normalizing, while computing $T\mathbf{1}$ at the same time and using it to normalize the output. Unfortunately, while this is mathematically sound, in the general case values skyrocket and overflow, causing numerical errors. We leave solving this numerical stability problem for future work.

## A.3  WEIGHT SHARING

While $A$ and $B$ play different roles, we could still expect them to be correlated. In this section we investigate the effect of a weight sharing of the form:

$$A = BD + D' \tag{9}$$

where $D$ and $D'$ are diagonal matrices. This is similar yet more general than Fagnou et al. (2024). Note that all the previous theoretical results still stand, since there was no assumption on $A$ and $B$ other than their sparsity pattern.

It turns out that the forward equation simplifies nicely:

$$\begin{aligned} y &:= (I - B)^{-1}(BD + D')x \\ &= (I - B)^{-1}(D + D')x - Dx \end{aligned} \tag{10}$$

Since addition and multiplication of diagonal matrices is linear and negligible, the only costly operation that remains is the multiplication by $(I - B)^{-1}$, and **the multiplication by $A$ disappeared**.

Looking now at the recursive form, by introducing the variable $z_i := y_i + d_i x_i$, we obtain:

$$z_i = \sum_{j=1}^{i-1} \beta_{ij} z_i + (d_i + d'_i)x_i \tag{11}$$

The recurrence got much simpler, and in particular the **cache size is reduced** since we only need to store the useful past $z_j$, instead of both the $x_j$ and $y_j$ in the general case.

## A.4 Efficient implementations

While implementing a custom CUDA kernel to perform the sparse triangular solves efficiently is out of the scope of this paper (we used the native torch function with is always quadratic) we discuss practical implementations in this section.

The causal token-mixing operators we study induce lower-triangular matrices $M \in \mathbb{R}^{n \times n}$ with sparse, repetitive structure. Forward substitution on such matrices has complexity proportional to the number of nonzeros. If each row has at most $w$ nonzeros, then solving $Mx = b$ requires $O(nw)$ operations, compared to $O(n^2)$ for dense triangular solves.

**Block forward substitution.** Because the sparsity patterns we consider are *periodic* along the diagonal (see Proposition 4.7), the system can be naturally partitioned into blocks, where each block shares the same nonzero structure. This allows the solve to be reorganized into a sequence of block updates:

$$x^{(k)} = M_{kk}^{-1} \Big( b^{(k)} - \sum_{\ell < k} M_{k\ell} x^{(\ell)} \Big),$$

where $M_{kk}$ denotes the $k$-th diagonal block. Each block update involves small dense solves (with identical shape across blocks), which can be vectorized and batched efficiently on GPUs.

**Parallelism.** While the numerical entries of $M$ vary, the periodicity ensures that the memory access pattern and dependency graph repeat exactly. This enables highly regular parallel implementations: a single kernel can encode the substitution pattern once, and apply it across all blocks with different coefficients. Such regularity improves cache efficiency and load balancing compared to generic sparse triangular solvers.

**Recursive structure.** If the sparsity pattern itself is recursive (e.g., defined hierarchically or via dilation), then one could further apply divide-and-conquer strategies, solving the system by recursively eliminating larger and larger blocks. This approach can reduce synchronization costs and naturally exposes parallelism across scales, complementing the block forward substitution scheme.

# B Examples encompassed by the framework

To make the framework concrete, we instantiate equations equation 1 and equation 2 on several familiar layers. The goal is not to commit to a particular parameterization of $A$ or $B$ (their structure is discussed later), but to show how widely used mixers drop out as simple choices of the *direct* channel $A$ and the *recursive* channel $B$. Read the second column as the per-token scalar recurrence; the last two columns display the corresponding matrix choices. This allows an apples-to-apples comparison of mechanisms that are usually presented as unrelated (attention vs. SSMs vs. hybrids), by expressing them in the same causal linear template.

| Layer | Scalar recurrence | $A$ | $B$ |
|---|---|---|---|
| Causal attention | $y_t = \sum_{j=1}^{t} \alpha_{tj} x_j$ | $A$ (row-stochastic) | 0 |
| Diagonal SSM | $y_t = a\, x_t + b\, y_{t-1}$ | $a\, I$ | strictly subdiagonal $b$ |
| Mamba / minGRU (scalar gates) | $y_t = a_t x_t + b_t y_{t-1}$ | $\mathrm{diag}(a_1, \ldots, a_L)$ | strictly subdiagonal $(b_t)$ |
| Conv + SSM block | *conv then SSM* | Toeplitz (from conv) | strictly lower |
| ChaCAL (damped, causal) | $y_t = (1-\gamma) \sum_{j \leq t} \alpha_{tj} x_j + \gamma \sum_{j < t} \alpha_{tj} y_j$ | $(1-\gamma)A$ | $\gamma\, (A \odot (\mathbf{1} - I))$ |

Table 3: Common layers as instances of the causal recurrent mixing template $\mathbf{y} = (I - B)^{-1} A\, \mathbf{x}$. "Direct" mixing lives in $A$; "recursive" coupling lives in $B$. For ChaCAL the diagonal of $A$ is zeroed to ensure strict causality and invertibility of $(I - B)$.

A few remarks help interpret the rows. *Causal attention* fits by setting $B = 0$, so all mixing is instantaneous and one-hop; $A$ is row-stochastic (softmax over scores) and provides global context in a single step. *Diagonal SSMs* and *minGRU/Mamba-style scalar-gated recurrences* correspond

to a strictly subdiagonal $B$ (one-step feedback) with either a constant or time-varying diagonal $A$; this yields linear-time decoding with a local, stepwise propagation of information. *Conv + SSM blocks* can be seen as a Toeplitz $A$ (the convolution) followed by a strictly lower $B$ (the recurrent accumulator), combining short-range pattern extraction with causal memory. Finally, *ChaCAL*-type rational mechanisms use $A$ twice – once as direct mixing and once inside the recursion – so that $(I - B)^{-1}A$ explicitly sums multi-hop paths; damping and a zero diagonal in $B$ keep the operator strictly causal and numerically well-behaved.

Beyond taxonomy, the table also signals design levers that we analyze elsewhere: sparsifying or banding $B$ controls hop depth and per-token work (triangular solves rather than dense matvecs), while choosing $A$ to be row-stochastic, diagonal, Toeplitz, or low-rank sets the one-hop mixing pattern and parameter-efficiency. In this sense, familiar architectures are not endpoints but particular coordinates in a larger space defined by the pair $(A, B)$.

# C    EXPERIMENTAL DETAILS

## C.1    DATASETS

**Synthetic tasks.**    All three synthetic datasets are generated on the fly during training, such that there is no overfitting problem. We employ some form of curriculum training as in (Dao & Gu, 2024a), with the training being split into 4 phases which divide the sequence length (and other task-specific parameters if suited) by respectively 8, 4, 2 and 1.

**Copy.**    The copy task is adapted from Arjovsky et al. (2016) and Jelassi et al. (2024). The model must copy sequences with length up to $L = 128$. The beginning and end of the input sequence are marked by special tokens.

**Associative recall.**    We adapt this task from Arora et al. (2024a). We use up to 64 key-value pairs, and sequences up to 256 long. We additionally randomize more the position of the keys and values to prevent any bias favoring a specific attention pattern.

**Multi-hop.**    We use the same setup as the associative recall task, but each value has a probability $p = 0.5$ to be replaced by a preceding key. Since the task is harder to learn, we also add labels to the intermediary keys to help the model learn.

**OpenWebText.**    This dataset was build to replicate the (undisclosed) training dataset of GPT-2 (Radford et al., 2019). It contains 38GB of text data from 8,013,769 documents. We use the same tokenizer as GPT-2.

## C.2    TRAINING SETUP

Training is performed on single NVIDIA V100 GPUs for the synthetic tasks, and pairs of NVIDIA A100 GPUs for language modeling. We use mixed precision with FP16.

All runs use a linear warmup for the learning rate, followed by a cosine scheduler.

## C.3    HYPERPARAMETERS

We report all hyperparameters in Table C.3

# D    PROOFS

## D.1    PROOF OF PROPOSITION 4.1

The result is relatively straightforward. At time $n$, we attend the past indices $n - f(0)$, $n - f(1)$, ..., $n - f(i)$, as long as $n - f(i) > 0$.

Table 4: Hyperparameters used in the different experiments.

| Name | Synthetic tasks | Language modeling |
|---|---|---|
| train steps | 20k | 100k |
| warmup steps | 2k | 4k |
| lr | 3e-3 | 4e-5 |
| batch size | $\geq 1024$ | 256 |
| weight decay | 0.1 | 0.1 |
| $\beta_1$ | 0.9 | 0.9 |
| $\beta_2$ | 0.98 | 0.98 |
| grad max norm | 1.0 | 1.0 |
| vocab size | 8,192 | 50,257 |
| context length | $\leq 256$ | 512 |
| num layers | 2 | 6 |
| dim | 256 | 512 |
| ff dim | 1024 | 2048 |
| head dim | 64 | 64 |

The number $k_n$ of past tokens attended is:

$$k_n := \#\{i \in \mathbb{N} \mid 0 < n - f(i)\} \tag{12}$$
$$= 1 + \max\{i \in \mathbb{N} \mid 0 < n - f(i)\} \tag{13}$$
$$= 1 + \max\{i \in \mathbb{N} \mid f(i) < n\} \tag{14}$$
$$= 1 + g(n) \tag{15}$$
$$= \mathcal{O}(g(n)) \tag{16}$$

If $f$ can be extended to an invertible real function $\mathbb{R} \to [1, \infty)$, then we have by definition of $g$ that:

$$f(g(n)) \leq n \tag{17}$$
$$\Longleftrightarrow g(n) \leq f^{-1}(n) \tag{18}$$

and hence $k_n = \mathcal{O}(f^{-1}(n))$.

## D.2 PROOF OF PROPOSITION 4.2

Given two positions $i < j$, we denote $d(i, j)$ the length of the shortest path from token $i$ to token $j$.

We can consider the underlying directed acyclic graph of the pattern: each token is a node, and there is an edge $i \to j$ iff $\exists k \in \mathbb{N}, f(k) = i - j$.

$$d(i, j) = \min\left\{k \mid \exists v \in [1, i]^{k-1}, i \to v_1 \to \cdots \to v_{k-1} \to j\right\} \tag{19}$$
$$= \min\left\{k \mid \exists v \in [1, i]^{k-1}, u \in \mathbb{N}^k, f(u_1) = v_1 - i, \ldots, f(u_k) = j - v_{k-1}\right\} \tag{20}$$
$$= \min\left\{k \mid \exists u \in \mathbb{N}^k, \sum_{p=1}^{k} f(u_p) = j - i\right\} \tag{21}$$

## D.3 PROOF OF COROLLARY 4.3

While the shortest path is a nontrivial quantity in the general case, we can find exact values for simple choices for $f$:

**Exponential** $f(i) = 2^i$: A key observation is that the shortest path does not involve two edges that share the same power of 2 – otherwise they could have been replaced by a single edge uses the next power of two. Hence we are looking for a way to decompose $j - i$ into a sum of *unique* powers of two. The only solution is given by its binary representation.

**Quadratic** $f(i) = (i+1)^2$: Lagrange's four-square theorem tells us that every natural number can be written as the sum of at most 4 squares. First, we can see that when $j - i \leq 3$ this is trivially true. Consider the number $m = j - i - 4$. By Lagrange's four-square theorem it can be written as $m = a^2 + b^2 + c^2 + d^2$. And hence $j - i = (a^2 + 1) + (b^2 + 1) + (c^2 + 1) + (d^2 + 1)$.

Note: while it is surprising to get a constant value, remind that exponential $f$ gives a logarithmic bound. Having a denser attention pattern should get a much better bound than logarithmic, which at our scale would appear constant.

### D.4    PROOF OF PROPOSITION 4.4

Suppose we have $n$ directed paths in a graph $\mathcal{G}$, each of length at least $d$ edges. Let the graph have $2n$ nodes (for the copy task setup). By definition, a path of length $d$ edges visits $d + 1$ nodes, so the total number of node visits across all paths is at least:

$$\text{total visits} \geq n \cdot (d + 1) \tag{22}$$

Let $C(\mathcal{G})$ denote the maximum number of paths passing through any single node. Since each node can be traversed by at most $C(\mathcal{G})$ paths, the total number of visits is also upper bounded by:

$$\text{total visits} \leq 2n \cdot C(\mathcal{G}) \tag{23}$$

Combining these inequalities, we obtain:

$$n \cdot (d + 1) \leq 2n \cdot C(\mathcal{G}) \tag{24}$$

$$\implies C(\mathcal{G}) \geq \frac{d + 1}{2} \tag{25}$$

### D.5    PROOF OF PROPOSITION 4.5

Consider $n$ paths of length at most $D$ edges, where the token mixing pattern is *translation-invariant*: the path for input $i + 1$ is a shift of the path for input $i$.

In this case, each node is visited by at most $D$ paths simultaneously, which occurs in the overlapping region of consecutive paths. Hence, the maximum congestion satisfies:

$$C(\mathcal{G}) \leq D \tag{26}$$

### D.6    PROOF OF PROPOSITION 4.7

If we note $p_i(t)$ the index we obtain by increasing $t - f(i)$ until reaching a cached token (with $t \geq f(i)$), we can write:

$$p_i(t) = \begin{cases} p_i(t-1) & \text{if } 0 < t - f(i) \leq p_i(t-1) \\ p_{i-1}(t-1) & \text{else.} \end{cases} \tag{27}$$

$$= p_{i-1}(f(i) - 1) + (p_{i-1}(f(i) - 1) + 1) \left\lfloor \frac{t - f(i)}{\underbrace{p_{i-1}(f(i) - 1) + 1}_{a_i}} \right\rfloor \tag{28}$$

$$= a_i \left( 1 + \left\lfloor \frac{t - f(i)}{a_i} \right\rfloor \right) - 1 \tag{29}$$

We can use this equation to find a recursive relation for $p_{i-1}(f(i) - 1)$:

$$a_i := p_{i-1}(f(i) - 1) + 1 \tag{30}$$

$$= a_{i-1} + a_{i-1} \left\lfloor \frac{f(i) - f(i-1) - 1}{a_{i-1}} \right\rfloor \tag{31}$$

$$= a_{i-1} \left( 1 + \left\lfloor \frac{f(i) - f(i-1) - 1}{a_{i-1}} \right\rfloor \right) \tag{32}$$

$$= \text{"the largest multiple of } a_{i-1} \text{ that is strictly greater than } f(i) - f(i-1) - 1\text{"} \tag{33}$$

$$= \text{"the largest multiple of } a_{i-1} \text{ that is greater or equal to } f(i) - f(i-1)\text{"} \tag{34}$$

$$= a_{i-1} \left\lceil \frac{f(i) - f(i-1)}{a_{i-1}} \right\rceil \tag{35}$$

## E  PYTHON CODE FOR GENERATING MATRICES

```python
def get_A(n, func, cache_efficient=False):
    A = torch.zeros((n, n), dtype=torch.bool)
    for i in range(n):
        A[i, i] = True

        for k in range(n):
            j = i - func(k)
            if j < 0:
                break

            if cache_efficient:
                # increase to closest next index in the cache
                while A[i-1, j] == 0:
                    j += 1

            A[i, j] = True
    return A
```