# OpenReview forum: "Trading Complexity for Expressivity: Theoretical Exploration of Linear and Causal Token Mixing Strategies"
_ICLR.cc/2026/Conference — ICLR 2026 Conference Withdrawn Submission_

### Official Review · Reviewer_Bkak · 2025-10-31

**Soundness:** 3
**Presentation:** 3
**Contribution:** 2
**Rating:** 2
**Confidence:** 5

**Summary:**

1. **Framework**: The paper introduces a framework to study sequence mixers that decomposes the mixer into two components: (i) one step influence, and (ii) past outputs influence.

    This is general enough to capture Attention and recurrent models like Mamba-2. With some simple math, the sequence mixer is equivalent to: $\mathbf{y} =  (\mathbf{I}-\mathbf{B})^{-1}\mathbf{A}\mathbf{x}$, where $\mathbf{B}$ is lower triangular matrix that encodes the locality/connections between tokens.

2. **Metrics**: They introduce two metrics, which they show are equivalent, namely Shortest Information path and Cogestion to quantify the distance between two interacting tokens, or equivalently the information flow through any token. They claim that the metrics capture a trade-off between expressivity and complexity in sequence mixers.

3. **Experiments**: The paper considers the following baselines: Attention, Local Attention, Local Recurrence, Quadratically-separated mixer, Exponentially-separated mixer and evaluate on (i) Language Modeling (ii) Copy (iii) Associative Recall (iv) Multi-hope recall

**Strengths:**

1. The paper is an easy read and is well written.
2. The framework is simple, interesting and novel.
3. The metrics chosen to quantify connected-ness/congestion are interesting and novel.

**Weaknesses:**

### Cache-Efficient Version

1. The proposed cache-efficient decoding modifies the recurrence pattern at inference is hacky and takes the model out of its training distribution.
2. The paper lacks thorough experiments evaluating whether this OOD inference behavior affects generalization. This needs to be validated across larger model scales (e.g., 750M–1.3B) and multiple downstream tasks.

### Language Modeling Experiments

1. The chosen context length of 512 is shorter than standard practice 2048-4096. This possibly limits the evaluation of long range dependencies.
2. The authors do NOT mention the model size. I estimate it to be around 6*(12*512*512) + 50257*512 = 44M. which is too small to draw meaningful conclusions. Results at larger scales (>= 350M) are needed to make credible claims.
3. For instance, authors report a 4-5 PPL gap between Transformers and SSD-like (one step recurrent) models, which is inconsistent with known results that SSD-like models are competitive with transformers. This could possibly be due to (i) insufficient tuning of hyperparameters like LR (ii) scale of the models chosen (iii) parameterization  chosen for A,B; authors must sweep amongst the best known parameterizations (e.g. SSD) (iv) Proper normalization and initialization of the new architectures (which authors left as future work). These sweeps need to be done to make the claim that “adding more connections improves performance”, which at this moment is not well supported.

### Synthetic Experiments

1. The issues (iii,iv) pointed out in (3) also apply to the synthetic experiments.
2. There are only minor gains in recall experiments. Given that $f(i)=i^2+1$ has a constant distance of 4 between any two tokens, I would expect almost 100% performance in recall.
3. It would be interesting to see how the models behave when the sequence length for the synthetics is scaled up.

### Lack of efficient kernels/latency evaluation
I understand that the authors mention that this is beyond the scope of this paper, but providing an efficient code for testing is important. I would recommend that the authors look into fast CUDA inverse solvers and report latency.

**Questions:**

1. It is unclear why the authors chose to train on 13B tokens for a 44M model. According to the Chinchilla scaling law, an optimal compute allocation would suggest training on roughly 900M?

2. Could the authors do a short comparison of their work against [1]? Both this work and Chimera use an inverse-operator for token mixing, but (i) have different motivations to derive this, and (ii) pursue different claims; Chimera generalizes SSMs across domains whereas this work focuses on improving LM capabilities.

---

[1]:  Chimera: State Space Models Beyond Sequences. Aakash Lahoti, Tanya Marwah, Ratish Puduppully, Albert Gu

---

### Official Review · Reviewer_sHBW · 2025-10-31

**Soundness:** 1
**Presentation:** 2
**Contribution:** 1
**Rating:** 2
**Confidence:** 4

**Summary:**

The paper proposes a unified operator-theoretic view of causal linear token mixing in sequence models. The authors claim that any causal linear mixer can be written as  $y=(I−B)^{−1}Ax$ where A represents direct input mixing and B represents recurrent structure. This yields a framework for reasoning about the trade-off between computational complexity and expressivity. Finally, the paper provides an empirical study to illustrate their results.

**Strengths:**

The paper is generally well-written, apart from some inconsitencies between sections (see weaknesses). The paper also investigates an interesting theoretical angle regarding expressivity and computational complexity. Finally, the empirical results are interesting, but it is hard to judge them given the framing of the paper.

**Weaknesses:**

The paper presents a theoretical framework to investigate expressivity vs. computational complexity, however this theoretical framework does either not capture any existing token mixing strategies (attention, SSMs, modern RNNs) or is entirely redundant (as explained below). Unfortunately, I'm unable to tell which of these two options is the case as the definition of equation (1) is ill-defined.
Conventially, a state-space model is written as two equations (state dynamics and output equation):
$ x_t = A_t x_{t-1} + B_t u_t,  y_t = C_t x_t $, with y the output and u the input.
Equation (1) is formulated differently, i.e., $ y = A x + B y $, with y the stacked outputs and x the stacked inputs (I'm aware that above state space equation is elementwise and this is stacked, we will come back to this in a second). If we take this at face value, this framework does only encompass existing token mixing strategies (attention, SSMs, modern RNNs) iff B=0 always, as no SSM feeds back its output in a layer (this is done in LSTMs, but not in any SSM). Therefore, any characterization of SSMs in the paper and especially in Appendix B is wrong. This is apparent in [1,2] that unify attention and SSMs in one framework and in the line of fast weight programmer (DeltaNet) work [3,4]. The other option is that eq. (1) is meant to model the state dynamics directly, i.e., would correspond to $x_t = A_t x_{t-1} + B_t u_t$, albeit in stacked form. If this is the case, the reformulation $x = (I - B)^{-1}Au$ is trivial and well-known, as it only rolls out the elementwise form $x_t = A_t x_{t-1} + B_t u_t$ into a stack form. In control theory this is the difference between open-loop and closed-loop, see e.g. [5,6], with the latter doing this roll-out explicitly in the appendix. In SSM research, this has also been noted multiple times, e.g. [2,7]. Therefore, the framework either does not accurately capture existing SSMs or the framework reduces directly to the standard formulation, which in both cases render the subsequent results questionable or irrelevant.

The results shown in Sec. 4, make no use of the framework, e.g., Prop. 4.1. makes use of function f, which is not defined and not connected to the dynamics (i.e. A, B) at all. Additionally, there is no discussion how these results should feature in the model design.
Given this divide, the connection of Sec. 4 to the main theory in Sec. 3 is unclear and therefore the empirical results are hard to assess.

[1] Dao & Gu (2024), "Transformers are SSMs: Generalized Models and Efficient Algorithms Through Structured State Space Duality", https://arxiv.org/abs/2405.21060

[2] Sieber et al. (2024), "Understanding the differences in foundation models: Attention, state space models, and recurrent neural networks", https://proceedings.neurips.cc/paper_files/paper/2024/file/f271a36160097fbdb06a9adeb1605343-Paper-Conference.pdf

[3] Schlag et al. (2021), "Linear Transformers Are Secretly Fast Weight Programmers", https://proceedings.mlr.press/v139/schlag21a/schlag21a.pdf

[4] Yang et al. (2024), "Gated Delta Networks: Improving Mamba2 with Delta Rule", https://arxiv.org/abs/2412.06464

[5] Anderson et al. (2019), "System Level Synthesis", https://arxiv.org/abs/1904.01634

[6] Sieber et al. (2021), "A System Level Approach to Tube-based Model Predictive Control", https://arxiv.org/abs/2103.02460

[7] Gu et al. (2022), "Efficiently Modeling Long Sequences with Structured State Spaces", https://arxiv.org/abs/2111.00396

**Questions:**

1. How are SSMs, e.g. Mamba-2, represented in the framework you present?
2. How does eq. (1) relate to the framework proposed in [2]?
3. In case, you are adding B as a new design parameter, i.e., adding some additional recurrences (this would partly alliviate my main point above) similar to LSTMs: How does this effect the implementation of your approach? Would it suffer the same drawbacks as LSTMs, which are inherently sequential in nature?

---

### Official Review · Reviewer_tbQc · 2025-11-01

**Soundness:** 2
**Presentation:** 2
**Contribution:** 2
**Rating:** 4
**Confidence:** 3

**Summary:**

This paper introduces a unified view of causal linear models in which any layer can be written as $y=(I−B)^{-1}Ax$. Specifically, it considers $A$ as an input mixer and $B$ as "multi-hop" propagation. The authors then proceed with designing simple and translation-invariant sparsity patterns on $B$ via a function $f$, which depends only on the position of tokens.
With that, the authors show a tradeoff between time and memory cache, and then propose add a cache-efficient variant to keep memory under the same cost as the time complexity.
Experiments with small language models and synthetic recall tasks showcase the impact of different pattens induced by $f$.

**Strengths:**

- The paper gives a neat linear view of causal token mixing, $y = (I - B)^{-1}Ax$. This can be a useful view to reason about the cost vs. expressivity tradeoff.

- By parameterizing recurrence with a simple position-dependent function $f$, the paper shows how to trade per-token time/cache for path length and congestion.

- The cache-efficient variant is interesting and can have practical impacts: it seeks to keep the "working set" small while maximizing "reachability" properties.

- Even if small, the synthetic experiments do test exactly the properties the theory talks about such as multi-hop recall and congestion.

**Weaknesses:**

- The framework proposed in this paper assumes that information will propagate along the induced fixed pattern from $f$ (the model is only given those edges), and thus the model is not actually "forced" to use all of them. That is, in practice, the network could fail to route information through the intended paths (e.g., sinking to just a few of them), which makes the "reachability" argument weaker empirically. I believe this can also make training harder or lead to suboptimal results. Further experiments with different model types (e.g., with SSMs, linear recurrent models, recurrent models, sparse transformers) would definitely strength the paper's main claims and allow readers to take more insightful conclusions.

- The complexity gains come from a fixed translation-invariant pattern. Of course, a learned or content-based sparsity would be more flexible but I guess that would break the clean theory and the possible tradeoffs.

- As noted by the authors, to actually leverage the time/cache cost induced by $f$, one needs specialized kernels that exploit the sparse pattern. It is not clear from the paper whether this can be done with existing attention implementations (e.g., FlexAttention) or whether each choice of $f$ would require its own kernel.

- I think the empirical section is too small given the ambition of the method. It covers tiny models (e.g., GPT-2), no long-context benchmarks, no comparison to strong linear recurrent baselines, and, perhaps more importantly, no wall-clock or memory benchmarks. As of now, it's hard to tell if the ideas in this paper can actually improve current sequential models.

**Questions:**

- Can you provide empirical evidence (e.g., via gradient/path tracing) that, during training, information about far tokens is in fact routed through the paths prescribed by $f$?

- What happens if the task's "true dependencies" do not align with the fixed pattern?

- Can the proposed mixers be implemented efficiently using existing flexible-attention frameworks (e.g. FlexAttention)?

---

### Official Review · Reviewer_zhwP · 2025-11-01

**Soundness:** 1
**Presentation:** 2
**Contribution:** 3
**Rating:** 2
**Confidence:** 2

**Summary:**

This paper proposes a theory connecting recurrant architectures (e.g. RNNs) with input "instantaneous" token-mixing architectures (e.g. Transformers). The framework is based on the following observation. Let $x$ be the input $y$ by the output, and consider the linear equation expressing $i$th output embedding as a function of the first $i$ input token embeddings and $i-1$ output token embeddings,

 - $y_i = \sum_{j=1}^i \alpha_{ij}x_j + \sum_{j=1}^{i-1} \beta_{ij}y_j$,

Consider stacking the embeddings to obtain matrices $X$, $Y$, and ordering the $\alpha,\beta$ coefficient into triangular matrices $A,B$ in the natural way. Then the previous equation is $Y = AX + BY$ and, since $B$ has a zero-diagonal, one can rearrange $Y = (I-B)^{-1}AX$.

From this observation, the authors theoretically examine different attention patterns, which I think correspond to different sparsity patterns on A and B. The authors examine how many recurrences (repeat applications of $(I-B)^{-1}A$?) it takes for a given token to affect another one. They present bounds on this "mixing" in terms of the time complexity for generating the $n$ token. The effect of caching is explored as well.

The paper also includes experiments on synthetic data on well-known tasks in the interpretability literature: copying the input, copying a part of the input corresponding to a given key, and a bounded recursion of this task.

**Strengths:**

- A clear model of deep NNs which places RNNs and transformers on tuneable spectrum is very appealing. As in any theoretical paper, the model in this paper makes simplifying assumptions (linearity) on deep NNs; nonetheless, if it were able to provide insightful theorems which then lead to compelling experimental results, this paper would be a significant contribution to the community.
- I have not seen the $Y= (I-B)^{-1}AX$ decomposition before, at least not in this form. In this regard the paper is original.

**Weaknesses:**

Despite significant effort, I was not able to parse most of the theoretical claims made in the paper. I suggest the authors look at examples of theoretical literature (e.g. the book of Shalev-Shwartz and Ben-David) for examples of what it means for a theory to be mathematically sound. It could be simply that I (a theoretical computer scientist) am not the taret audience for this paper; I acknowledge that NeurIPS has multiple theory communities in it, and I am giving this review a low confidence score to account for this event.

- With theoretical work, it is crucial to clearly explain what the assumptions being made are, for all the results (should) rely on these assumptions. These assumptions are typically given in a "Definition" environment containing formal mathematics. These were missing from Section 3.
- A and B do not explicitly appear in any of the propositions or corollaries. It would be helpful to explain how Section 4 relates to Section 3, because right now (despite trying) I could not understand the formal connection.
- Is Proposition 4.2 actually a definition? Currently, equation (3) therein appears to claim that the "distance" between (i,j) is equal to the expression on the right hand side. However, it is not clear what the _left hand side_ is referring to. Is there some graph or other metric space to which $d$ is referring to? What is a path between tokens?
- Similarly, in section 4.1.2 $\mathcal{G}$ is not defined and I do not know what it is.
- Since the domain of $f$ is finite length strings over natural numbers (which is the standard interpretation of $\mathcal{N}^*$), none of its uses are well defined. What is $t - f(0)$ when $f(0)$ is a string of natural numbers?
- I could not understand Section 4.2

In particular, I would suggest move Section 3.2 into an "extended related work" appendix, it seems to be an advanced discussion of comparison to specific literature. Instead, for the benefit of most readers who aren't specialists in this line of work, use this valuable space early in the paper to formally define the objects used in the claims that follow. In addition, I suggest including concrete examples early on, before presenting a general theory (e.g. present the case of $f(i) = 2^i$ and concretely explain what ML setting this captures, and what your theorem is saying about this setting.)

**Questions:**

- What is a path between tokens?
- What is the distance in Equation 3?
- How is the graph $\mathcal{G}$ formally defined?
- What is the formal definition of "the cache efficient pattern induced by $f$"?

---

### Note · Authors · 2025-12-03

**Comment:**

We thank all reviewers for their thoughtful comments and suggestions. After careful consideration, we have decided to withdraw our submission from ICLR. Although we have made substantial progress in improving the paper since the initial version -- and are excited about its current direction -- we feel that, given the circumstances, continuing in this review cycle is not the best fit. The reviewers' feedback has been genuinely helpful in sharpening the clarity of the paper and strengthening the experimental design, and we are sincerely grateful for the time and thought they put into their comments. Unfortunately, the recent leak of the reviews has also complicated the discussion phase. We will take the time to further develop the work and present it in its best form going forward.

**Withdrawal Confirmation:**

I have read and agree with the venue's withdrawal policy on behalf of myself and my co-authors.